# One-Year Follow-Up Cognitive Decline After Hip Fracture Surgery: The Prognostic Role of NSE and S100B Biomarkers in Elderly Patients, a Multicentric Study

**DOI:** 10.3390/jfmk10040380

**Published:** 2025-10-01

**Authors:** Michele Coviello, Delia Barone, Antonella Abate, Alessandro Geronimo, Giuseppe Danilo Cassano, Vincenzo Caiaffa, Giuseppe Solarino, Giuseppe Maccagnano

**Affiliations:** 1Orthopaedics Unit, Department of Clinical and Experimental Medicine, Faculty of Medicine and Surgery, University of Foggia, Policlinico Riuniti di Foggia, 71122 Foggia, Italy; 2Interdisciplinary Department of Medicine, Pediatric Section, Children’s Hospital Giovanni XXIII, University of Bari “Aldo Moro”, 70126 Bari, Italy; deliabarone91@gmail.com; 3Orthopaedic and Traumatology Unit, “Di Venere” Hospital, 70131 Bari, Italy; antonella.abate@asl.bari.it (A.A.);; 4Orthopaedic and Trauma Unit, Department of Basic Medical Sciences, Neuroscience and Sense Organs, School of Medicine, AOU Consorziale Policlinico, University of Bari “Aldo Moro”, Piazza Giulio Cesare 11, 70124 Bari, Italy

**Keywords:** postoperative cognitive dysfunction, POCD, neuron-specific enolase, NSE, S100B protein, femur fracture

## Abstract

**Background:** Postoperative cognitive dysfunction (POCD) is a prevalent complication in elderly patients undergoing hip fracture surgery, often resulting in increased morbidity and prolonged rehabilitation. Biomarkers such as Neuron-Specific Enolase (NSE) and S100B protein have shown potential in detecting cerebral injury, yet their role in predicting long-term cognitive decline remains unclear. This study aimed to evaluate the association between biomarkers serum levels and the incidence of POCD in elderly patients undergoing proximal femur fracture surgery. **Methods:** A multicentric prospective observational study was conducted from January 2023 to February 2024, including 146 elderly patients with hip fractures treated surgically at ASL Bari and the University Orthopedic Department of Foggia. Biomarker levels of NSE and S100B were measured preoperatively (T0), at three days post-surgery (T1), and at one-year follow-up (T2). Cognitive function was assessed using the Pfeiffer Scale (PS) and the Mini-Mental State Examination (MMSE). Statistical analysis was performed using U Mann–Whitney tests and logistic regression to identify risk factors. **Results:** At three days post-surgery, 20.5% of patients exhibited POCD, with no significant differences in NSE and S100B levels compared to baseline. However, at one year, of the 96 patients investigated 37.9% of patients showed cognitive decline, with significantly elevated NSE (19.88 ± 4.03 μg/L) and S100B (1.86 ± 0.9 μg/L) compared to non-POCD patients (*p* = 0.01). Risk factors for long-term POCD included older age (OR: 1.24), diabetes mellitus (OR: 4.41), and lower baseline cognitive function (MMSE and PS scores, OR: 0.25 and 9.81, respectively). **Conclusions:** The study demonstrates that while early POCD is not associated with significant changes in NSE and S100B levels, their elevation at one-year follow-up suggests a possible correlation with chronic neuroinflammation and persistent neuronal damage. Preoperative cognitive impairment, advanced age, and diabetes mellitus are significant predictors of long-term cognitive decline. Incorporating biomarker evaluation and cognitive screening into perioperative management may enhance patient outcomes following hip fracture surgery.

## 1. Introduction

Postoperative cognitive dysfunction (POCD) is a frequent complication after surgery, with incidence rates ranging from 8.9% to 46.1%. Its pathogenesis is multifactorial, involving neuroinflammation, oxidative stress, neuronal injury, and impaired neurotrophic support, but remains incompletely understood [1].

A significant portion of an orthopedic surgeon’s clinical activity is dedicated to the care of elderly patients. This is particularly evident in the management of proximal femur fractures, where the majority of affected individuals are over 75 years of age. The advanced age of this patient population not only increases the risk of postoperative complications but also makes them particularly vulnerable to cognitive impairments such as postoperative cognitive dysfunction (POCD) and delirium, leading to increased morbidity, prolonged hospital stays, and a decline in overall quality of life [1,2].

One of the factors that has been shown to significantly reduce complications, including delirium, in patients with hip fractures is adherence to the guideline recommending surgery within 48 h. Even in patients taking direct oral anticoagulants (DOACs)—a factor that often delays surgery—timely operative management is associated with shorter hospital stays, fewer medical and surgical complications, and potential benefits in cognitive outcomes [3].

While neuropsychological testing is currently the mainstay for diagnosis, these assessments are limited by cultural and educational variability. They represent a useful approach for POCD assessment [4]. Moreover, greater attention should be directed toward blood biomarkers, which may provide a more reliable and objective method for predicting cognitive decline. Molecules such as amyloid beta (Aβ), tau, and neurofilament light (NFL) are well established in neurodegenerative diseases, while S100β, neuron-specific enolase (NSE), and glial fibrillary acidic protein (GFAP) have been linked to brain injury in stroke and trauma [5].

Neuron-Specific Enolase (NSE) and S100B protein levels may be slightly elevated in elderly individuals as a result of age-related changes, increased blood–brain barrier permeability, and comorbidities. NSE is a glycolytic enzyme predominantly expressed in neurons, and its serum elevation reflects neuronal injury and has been linked to neurodegenerative processes. S100B, a calcium-binding protein secreted by astrocytes, contributes to neuroinflammation and is associated with blood–brain barrier dysfunction and cognitive decline [6]. Pronounced increases in both biomarkers are more commonly observed in pathological conditions such as traumatic brain injury, stroke, dementia, or renal impairment. In studies on postoperative cognitive dysfunction, NSE and S100B levels rise significantly after surgery compared with non-operated elderly controls, suggesting that surgical and anesthetic stress exacerbate neuronal injury beyond normal age-related variability [4,7].

Recent studies indicate that elevated serum levels of NSE and S100B in elderly hip fracture patients correlate with higher risk of POCD and postoperative delirium, particularly when combined with inflammation [8,9]. Furthermore, anesthesia type, intraoperative hypotension, and systemic inflammatory responses contribute to perioperative neurocognitive disorders (PND), making it essential to explore the pathophysiological mechanisms underlying POCD [10].

Although increasing evidence suggests that NSE and S100B may serve as predictive markers of postoperative cognitive decline, their role and diagnostic accuracy in orthopedic trauma patients remain uncertain. This multicenter prospective study evaluated the association between serum NSE and S100B levels and cognitive outcomes in elderly patients undergoing femoral fracture surgery. We found that both biomarkers independently predicted long-term cognitive decline one year after surgery, extending previous short-term findings and linking their elevation to chronic neuroinflammation and neuronal injury. In addition, advanced age, diabetes mellitus, and poor baseline cognitive function emerged as independent risk factors. These results support the integration of biomarker assessment with cognitive screening to improve perioperative risk stratification and monitoring of postoperative cognitive dysfunction.

## 2. Materials and Methods

### 2.1. Patients and Study Design

A multicentric prospective observational study was conducted on consecutive patients who had surgery for proximal femur fractures. The study included patients admitted to Orthopaedic and Traumatology Units of ASL Bari Department and Orthopaedic Department of Foggia University between January 2023 and February 2024.

The study was conducted in accordance with the guidelines of the Declaration of Helsinki and approved by the Local Ethics Committee. Informed consent was obtained from the patients. The sample size for a prospective study was estimated using a S-100B serum level, with previous published values on the same treatment protocol [11] serving as the primary endpoint. Using a standard deviation of 0,05 ug/L, we predicted that 84 participants would be required to detect statistically significant differences at an α level of 0.05 and power level of 95%.

The inclusion criteria were: age greater than 65 years, proximal femur fracture (AO Classification from 31-A1.1 to 31-A3) [12] treated with proximal femur nail, cannulated screws and arthroplasty replacement, no history of neurological impairment and visual and auditory impairments. The exclusion criteria were: pathological fracture, open fracture, hematological haemolytic diseases, subject refuses surgical treatment or unable to complete the cognitive function, persistent neurological deficits as a result of stroke or shock due to trauma, psychiatric diseases, acute or chronic acute renal and liver function failure [4,5,6,7].

To determine the type of fracture, pre-operative radiographs in anterior–posterior and lateral views were analyzed and screened by two independent studies (G.M. and V.C.). In case of disagreement, a consensus meeting was held, and a third researcher (D.B.) was consulted to reach a final determination. According to the Orthopedic Trauma Association (AO/OTA) classification, fractures were separated into two groups: extracapsular, operated by osteosynthesis, and intracapsular, treated by replacement or cannulated screws. According to our department′s procedures, the patients had surgery within two days of their admission [12]. Low-molecular-weight heparin (LMWH) was administered prophylactically to all patients. Within 30 min of the procedure, a single dose of antibiotics was given. All patients were operated on by surgeons with at least five years of experience in the field of traumatology. A standard procedure required giving patients 1 g of acetaminophen as premedication two hours before surgery. Patients received spinal anesthesia while undergoing surgery performed by the same anesthesiologic team group. Three-channel electrocardiography, end-tidal carbon dioxide partial pressure, oscillometric blood pressure, and pulse oximetry were all tracked as standard procedure [13].

Blood samples were obtained before anesthesia induction, as well as the established follow-ups. After centrifugation at 2150× *g* for 10 min, serum samples were stored at −80 °C for future use. The serum levels of S-100B protein were determined using an enzyme-linked immunosorbent test (Biosource, Camarillo, CA, USA) as per the manufacturer′s instructions to ensure accuracy and reproducibility. The intra- and inter-assay coefficients of variation (CV) for the ELISA method were 4.5% and 7.8%, respectively, demonstrating high assay reliability. The NSE values were determined using a sandwich approach with NSE kits and the Elecsys 2010 Immunoassay analyzer (Roche Diagnostics, Vienna, Austria). The intra- and inter-assay coefficients of variation for the Elecsys NSE assay were 2.3% and 5.1%, following the recommended operational procedures to maintain consistency across tests [14].

Cognitive impairment was measured by Pfeiffer scale (PS) and Mini-Mental State Examination (MMSE). The first questionnaire collects patient errors from 10 basic questions and defines cognitive decline into four groups based on intellectual dependency. 0–2 errors indicate no cognitive decline, 3–4 errors indicate minor impairment and need for assistance with complex tasks, 5–7 errors indicate moderate deterioration requiring occasional assistance, and 8–10 errors indicate severe deterioration requiring ongoing supervision. We consider POCD if moderate to severe result [15]. The mini-mental state examination examined six primary areas: time, location orientation, expression and mathematical abilities, attention, and memory. The total score was 30 points, including the full evaluation and educational condition. A score below 20 points indicates medium to high cognitive dysfunction [16]. Even one positive test is sufficient to diagnose POCD.

The following data were recorded for each patient: age, sex, BMI, side of surgery, American Society of Anesthesiologists score (ASA) [17], surgical time, fracture type classification, hypertension, history of smoking, hypercholesterolemia, diabetes mellitus, peripheral vascular disease, history of myocardial infarct, NSE and S-100B serum levels, MMSE and PS questionnaires.

Data were collected at the following times: T0 (before the surgical procedure); T1 (3 days post-surgery); T2 (1-year post-surgery). Prior to enrollment, the study team should emphasize to participants the importance of long-term follow-up and the potential benefits of staying in the study. Providing clear, concise explanations about the study′s goals and the necessity of the follow-up would help ensure commitment to minimizing loss in follow-up.

A total of 177 patients meeting the inclusion criteria were included in the study.

Three patients declined to participate, and 28 subjects had positive diagnosis of POCD on testing before surgery. Therefore, they were excluded from the study. Subsequently, 146 patients were included and 95 patients were checked at 1 year follow-up. An overview of the number of patients recruited, enrolled and analyzed in this work is presented in Figure 1.

The primary endpoint of the study was the association assessment between cognitive impairment and biomarkers serological increase. The secondary endpoint was to determine the risk factors for the cognitive decline development.

### 2.2. Statistical Analysis

Statistical analyses were executed out using SPSS for Windows software (version 17.0; SPSS Inc., Chicago, IL, USA). Descriptive statistics for the full sample were calculated. Categorical data were represented as percentages and numbers, whilst continuous variables were represented as mean and standard deviation (SD). Non-parametric tests were performed due to the non-homogeneous distribution of the values, which was validated by the Kolmogorov–Smirnov test (*p* > 0.05).

The U Mann–Whitney test was employed to investigate the continuous parameters, and the risk factors were assessed using binary logistic regression. Logistic regression analysis was conducted to explore predictors of POCD at 1 year. After univariate testing, variables with *p* < 0.10 were entered into a multivariate logistic regression model. To assess model validity, we performed the Hosmer–Lemeshow goodness-of-fit test and calculated adjusted odds ratios (ORs) with 95% confidence intervals [18].

A *p*-value of less than 0.05 was taken to indicate significance.

## 3. Results

From January 2023 to February 2024, 146 individuals had hip fracture surgery. Table 1 shows the demographic and preoperative characteristics of all enrolled individuals. There were 88 females (60.2%). The mean age was 83.17 ± 8.21 years. Ninety-six patients (65.8%) were diagnosed with an extracapsular fracture, and 89 (60.9%) subjects were categorized as having more than three ASA grades.

At three days post-surgery, 30 patients (20.5%) exhibited POCD based on test results. Our analysis showed no statistically significant differences in biomarker levels between two groups regarding recruitment and the third day. The baseline values of the neuropsychological tests were significant, as presented in Table 2.

Ninety-five patients completed the study after one year of follow-up. Our analysis revealed that the two biomarkers were significantly elevated in patients who developed cognitive impairment compared to neurologically healthy individuals. Additionally, the results of the two neuropsychological tests showed a statistically significant association with cognitive impairment. These findings are summarized in Table 3.

To focus on possible risk factors for cognitive impairment after one year of follow-up among preoperative characteristics, a binary logistic regression was conducted with adjusted variables and summarized in Table 4. Our data reported a correlation for older age, the presence of diabetes mellitus, a poorer value of neuropsychological questionnaires. To address potential overlap between MMSE and PS, we evaluated multicollinearity using tolerance and variance inflation factors (VIF). Both variables demonstrated acceptable tolerance values (>0.6) and VIF < 2, indicating no significant collinearity. Therefore, both were retained in the model as independent predictors.

The increase in biomarkers from recruitment to one year of follow-up was graphically represented in Figure 2. As shown in Table 3, their difference is statistically significant.

## 4. Discussion

Postoperative cognitive dysfunction (POCD) remains a prevalent and clinically significant complication following hip fracture surgery, particularly in elderly patients. The findings from our study suggest that while short-term POCD (3 days post-surgery) was not associated with significant changes in NSE and S100B levels, these biomarkers were markedly elevated in patients who developed cognitive impairment at one-year follow-up. These results reinforce the hypothesis that long-term cognitive decline in orthopedic trauma patients may be linked to persistent neuroinflammation and neuronal damage [14,19]

The contrasting trends of NSE and S100B levels in short-term vs. long-term postoperative cognitive dysfunction (POCD) highlight key pathophysiological mechanisms underlying cognitive impairment in hip fracture patients. While NSE and S100B levels did not show significant changes at 3 days postoperatively, their elevation at 1 year was strongly correlated with persistent cognitive decline. This discrepancy suggests as in previous papers that early POCD is more likely driven by transient factors, such as anesthesia effects and perioperative stress, while long-term cognitive impairment reflects chronic neurodegenerative process [20,21].

The lack of significant NSE and S100B elevations in early POCD suggests that acute neuronal damage is not the primary driver of cognitive dysfunction in the immediate postoperative period. The following are some potential reasons for early factors of POCD. General anesthesia has been linked to transient cognitive impairment and neuroinflammation, with studies showing a temporary disruption in synaptic plasticity and cholinergic pathways [22]. Systemic inflammation following hip fracture surgery can alter blood–brain barrier (BBB) integrity, leading to neuroinflammatory cascades without immediate neuronal injury [23]. Intraoperative hypotension and microembolization during orthopedic procedures may contribute to transient cognitive decline, although these effects are often reversible [24]. At present, these studies do not suggest any certainties, so the matter is still under investigation. In addition, a recent meta-analysis review also adds details regarding the choice of anesthetic drugs for surgery, opening up scenarios of extreme research complexity. Moreover, several confounding factors, such as adherence to post-operative rehabilitation or the use of medications (sleeping pills or painkillers), could influence long-term POCD, which is still being researched [25].

Conversely, the significant increase in NSE and S100B levels at 1-year follow-up suggests an ongoing neurodegenerative process contributing to cognitive decline. This aligns with recent findings that elevated NSE levels correlate with long-term cognitive impairment in elderly surgical patients [21]. The persistent elevation of S100B at 1 year suggests prolonged glial activation, which has been implicated in neurodegenerative diseases such as Alzheimer′s and vascular dementia [26]. NSE elevation at 1 year may indicate delayed neuronal injury caused by oxidative stress, mitochondrial dysfunction, and chronic neuroinflammation [27]. Cumulative effects of metabolic dysregulation: Diabetes mellitus and other metabolic disorders may exacerbate neuronal vulnerability, leading to progressive cognitive decline post-surgery [28].

Altogether, these findings suggest that while early POCD could be reversible, long-term cognitive impairment is more closely linked to chronic neuroinflammation and sustained neuronal damage. NSE and S100B may therefore serve as valuable prognostic biomarkers for long-term POCD risk assessment rather than short-term diagnostic tools. Given the delayed elevation of NSE and S100B in POCD patients, their utility as early diagnostic markers for short-term POCD may be limited. However, their significant increase at 1-year follow-up suggests a role in long-term cognitive monitoring. Future studies are necessary to optimize the sampling time point for the better prediction of POCD. They are warranted to unify the source of biomarkers, sampling time points, and diagnostic methods of POCD so as to further demonstrate the predictive value of biomarkers in perioperative cognitive function [29].

Based on these preliminary results, certain subpopulations of patients at higher risk for postoperative cognitive decline could then be selected to undergo additional investigative approaches. Zohu et al. propose structural and functional MRI associated with arterial spin labeling (ASL) to evaluate cortical gray matter volumetry and cerebral perfusion in the early warning of POCD to elderly abdominal surgical patients. Due to high costs and logistical constraints, these imaging techniques are not feasible for routine use in all patients and are generally reserved for selected high-risk individuals [30].

Our logistic regression analysis revealed several independent risk factors for 1-year POCD, including older age, diabetes mellitus, and preoperative cognitive impairment. These findings are consistent with prior studies emphasizing the role of vascular dysfunction and metabolic factors in exacerbating cognitive decline following surgery [30,31]. Older patients exhibited a significantly higher risk of POCD (OR: 1.24, *p* = 0.01), likely due to reduced neuroplasticity, cerebrovascular changes, and a heightened inflammatory response post-surgery [32]. The four-fold increase in POCD risk in diabetic patients (OR: 4.41, *p* = 0.01) underscores the impact of metabolic dysregulation, insulin resistance, and microvascular complications on brain function [31]. A comprehensive meta-analysis encompassing 38 trials (*n* = 8.748) showed that diabetic patients have a significantly increased risk of POCD compared to non-diabetics (OR 1.44), particularly in those older than 65 years [33]. A recent review including over 67.000 subjects reinforced that older age and preoperative diabetes are among the most consistent risk factors for POCD across various surgical types, particularly in cardiac surgery [34]. A range of other patient-related factors—including obesity, metabolic disorders, and nutritional status—have been shown to affect the risk and severity of postoperative cognitive dysfunction (POCD). As highlighted by recent evidence in cardiac surgery patients, malnutrition is associated with increased inflammation, post-operative complications, and a higher risk of postoperative cognitive dysfunction (POCD). Therefore, incorporating the assessment and optimization of nutritional status into the perioperative management of fracture patients may contribute to reducing complications and mitigating the risk of POCD [35]

Patients with lower MMSE scores and higher Pfeiffer Scale values at recruitment were significantly more likely to develop POCD, emphasizing the importance of baseline cognitive screening in orthopedic patients [36]. Lower baseline cognitive scores, as measured by tools like the MMSE and Pfeiffer Scale, are associated with a greater risk of postoperative cognitive dysfunction (POCD) in hip fracture patients, which suggests that conducting routine cognitive assessments may be useful for identifying at-risk individuals.

The study′s findings are limited by the absence of data on key inflammatory and oxidative stress markers, which are increasingly recognized as central to the development of POCD. Future studies should address this gap by including these markers. Additionally, exploring the impact of different anesthesia types and surgical methods on POCD is crucial. This could be achieved through larger-scale, retrospective studies that analyze existing patient data to identify specific anesthetic agents or surgical techniques that may be associated with a higher or lower risk of POCD. By addressing these limitations and expanding the scope of research, future studies can move toward developing more accurate prognostic models and ultimately, better preventive and therapeutic strategies for POCD in orthopedic trauma patients.

## 5. Conclusions

This study demonstrates that while NSE and S100B levels are not significantly altered in early POCD, their elevation at one year is strongly correlated with persistent cognitive impairment. Older age, diabetes, and low baseline cognitive scores emerged as key risk factors, underscoring the importance of preoperative screening and postoperative monitoring. In this context, NSE and S100B should be regarded as prognostic rather than diagnostic tools, with targeted use in high-risk patients rather than universal application, given the lack of standardized cut-offs and additional costs. Selective preoperative and follow-up testing, particularly at one year, may help identify patients who would benefit from intensified cognitive surveillance, optimization of vascular and metabolic risk factors, and early neurogeriatric referral. Future research should focus on assay harmonization, cost-effectiveness, and interventional trials to establish whether biomarker-guided care can effectively reduce long-term cognitive decline.

## Figures and Tables

**Figure 1 jfmk-10-00380-f001:**
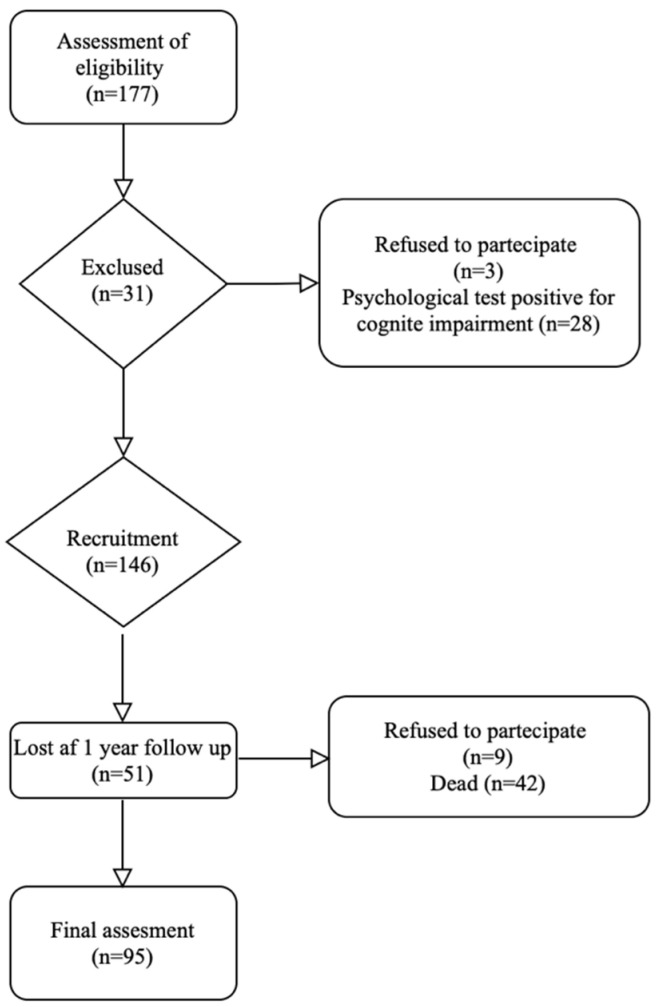
Diagram of the number of patients enrolled and analyzed in this study.

**Figure 2 jfmk-10-00380-f002:**
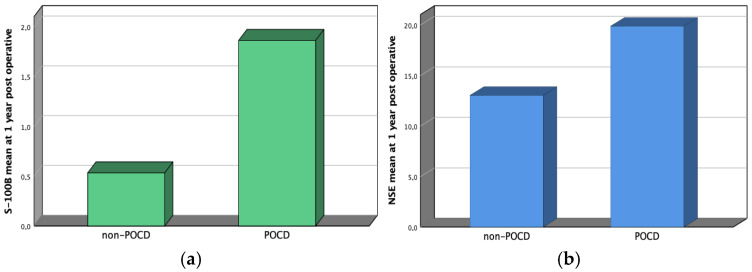
Graphical representation of the two biomarkers ((**a**): S-100B; (**b**): NSE) at one-year follow-up among cognitive impairment and noncognitive impairment patients.

**Table 1 jfmk-10-00380-t001:** Main data of the study.

Preoperative Features	(*n* = 146)
**Age (year)**	83.17 ± 8.21
**Gender (female)**	88 (60.2%)
**BMI (Kg/m ^2^)**	27.30 ± 4.92
**Side (right)**	69 (47.3%)
**Type of fracture**	
Intracapsular	50 (34.2%)
Extracapsular	96 (65.8%)
**Surgical Time (min)**	25.17 ± 9.91
**ASA**	
≤2	57 (39.1%)
>3	89 (60.9%)
**Hypertension**	65 (44.5%)
**History of smoking**	69 (47.3%)
**Hypercholesterolemia**	60 (41.1%)
**Diabetes mellitus**	27 (18.4%)
**Peripheral vascular disease**	5 (3.4%)
**History of myocardial infarct**	5 (3.4%)
**NSE (μg/L)**	15.41 ± 4.90
**S-100B (μg/L)**	0.51 ± 0.22

(Data are presented as mean ± standard deviation or number and percentage; BMI: Body Mass Index; NSE: neuron-specific enolase: S-100B: plasma S-100B protein).

**Table 2 jfmk-10-00380-t002:** Differences in POCD versus non-POCD groups at 3 days post.

	POCD *n* = 30 (20.5%)	Non-POCD *n* = 116 (79.5%)	*p*-Value
**NSE at recruitment**	15.88 ± 5.47	15.27 ± 4.86	0.31
**NSE 3d**	16.57 ± 5.63	15.91 ± 5.18	0.33
**S-100B at recruitment**	0.49 ± 0.21	0.52 ± 0.22	0.35
**S-100B 3d**	0.56 ± 0.15	0.52 ± 0.21	0.65
**MMSE at recruitment**	18.37 ± 5.89	21.97 ± 5.08	**0.04**
**PS at recruitment**	4.97 ± 0.28	3.53 ± 0.69	**0.02**

(One hundred and forty-six patients; U Mann–Whitney test; data are presented as mean ± standard deviation; POCD: post-operative cognitive dysfunction; NSE: neuron-specific enolase: S-100B: plasma S-100B protein; 3d: 3 days post operative; MMSE: mini-mental state examination; PS: Pfeiffer scale).

**Table 3 jfmk-10-00380-t003:** Differences in POCD versus non-POCD groups at 1 year follow-up.

	POCD *n* = 36 (37.9%)	Non-POCD *n* = 59 (62.1%)	*p*-Value
**NSE at recruitment**	15.28 ± 6.06	15.85 ± 4.81	0.82
**NSE 3d**	15.71 ± 5.07	15.76 ± 5.18	0.76
**NSE 1y**	19.88 ± 4.03	16.69 ± 4.67	**0.01**
**S-100B at recruitment**	0.51 ± 0.24	0.53 ± 0.19	0.89
**S-100B 3d**	0.54 ± 0.23	0.55 ± 0.31	0.77
**S-100B 1y**	1.86 ± 0.9	0.78 ± 0.45	**0.01**
**MMSE at recruitment**	15.01 ± 2.64	24.76 ± 4.87	**0.01**
**PS at recruitment**	3.94 ± 0.23	3.46 ± 0.72	**0.03**

(Ninety-five patients; U Mann–Whitney test; data are presented as mean ± standard deviation; POCD: post-operative cognitive dysfunction; NSE: neuron-specific enolase: S-100B: plasma S-100B protein; 3d: 3 days post operative; 1y: 1 year post operative; MMSE: mini-mental state examination; PS: Pfeiffer scale).

**Table 4 jfmk-10-00380-t004:** Binary logistic regression analysis for predictors of postoperative cognitive dysfunction at 1 year, adjusted for age, diabetes mellitus, MMSE, and PS.

Factor	*p*-Value	OR	95% CI for OR
**Age**	**0.01**	1.24	1.13–1.34
**Diabetes mellitus**	**0.01**	4.41	1.48–13.16
**MMSE at recruitment**	**0.01**	0.25	0.10–0.60
**PS at recruitment**	**0.01**	9.81	2.27–42.49

(MMSE: mini-mental state examination; PS: Pfeiffer scale).

## Data Availability

Data are unavailable due to privacy or ethical restrictions.

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
