# Peer review of "One-Year Follow-Up Cognitive Decline After Hip Fracture Surgery: The Prognostic Role of NSE and S100B Biomarkers in Elderly Patients, a Multicentric Study"

_jfmk, 2025, doi:10.3390/jfmk10040380_

Round 1

Reviewer 1 Report

Comments and Suggestions for Authors

Thank you to the authors for this work.
Thank you to the editor for this review opportunity.
This is a clear, well-presented article on an interesting topic.
The context is well-introduced.
However, here are my comments/questions:

You compare patients who have undergone surgery and examined their cognitive decline as well as their biomarker levels in a predominantly elderly population. Is it possible to compare your figures to the general population for age? In other words, what do the elevated biomarker levels represent compared to a non-operated population whose cognitive decline is nonetheless undoubtedly present at an advanced age? Are these levels then significant?
You specify that recent studies have shown a correlation between elevated NSE and S100B levels and a risk of cognitive decline and postoperative delirium, see references 8 and 9. What does your study add compared to these two references?
I don't really understand your statement about the surgeon and his 5 years of experience. Furthermore, were all the procedures actually performed by the same surgeon? Please rephrase/explain and clarify.
You report that your data demonstrated a correlation between advanced age, diabetes, and poorer performance on neuropsychological questionnaires. Is this consistent with the existing literature? Can you correlate these results with the literature? Please clarify and elaborate.
Excellent discussion; very clear and relevant.
You suggest that NSE and S100B may be valuable biomarkers for assessing the risk of postoperative cognitive decline. What might the strategy be in this regard? What would be your advice or recommendations for the future? Do you recommend systematic screening of these biomarker levels preoperatively and postoperatively to detect neurological decline earlier? What do you know about the cost of these analyses? Could they be systematically added to the preoperative assessment before orthopedic surgery? Before other surgeries? Is this easily achievable by most laboratories? Is the cost worth the analysis?
How can we organize a strategy using these biomarkers to truly reduce neurological decline in clinical practice? Could you explain, rephrase, or provide details in the text?
Thank you.

Author Response

Semptember 22, 2025

Journal of Functional Morphology and Kinesiology

Reviewers

& c.c.

Manuscript Editor

Dear Editor,

            Thank you for your thoughtful and constructive comments. In this cover letter, we have addressed each of the issues raised and have highlighted the relevant revisions in the manuscript itself (underlined). Below, please find item-by-item responses to the Reviewers’ comments.

Please note: Editor’s and Reviewers’ comments are in italicized red font; Authors’ answers are in regular black font;

Sincerely yours,

Michele Coviello, MD PhD student

Orthopaedics Unit, Department of Clinical and Experimental Medicine, Faculty of Medicine and Surgery, University of Foggia, Policlinico Riuniti di Foggia, 71122 Foggia, Italy

E-mail: michelecoviello91@gmail.com, ORCID: 0000-0003-3585-1000, Phone: +393938165088

Reviewer 1

Thank you to the authors for this work.

Thank you to the editor for this review opportunity.

This is a clear, well-presented article on an interesting topic.

The context is well-introduced.

We thank the reviewer for feedback.

However, here are my comments/questions:

You compare patients who have undergone surgery and examined their cognitive decline as well as their biomarker levels in a predominantly elderly population.

Is it possible to compare your figures to the general population for age? In other words, what do the elevated biomarker levels represent compared to a non-operated population whose cognitive decline is nonetheless undoubtedly present at an advanced age? Are these levels then significant?

We improve introduction with section comparing cognitive decline in general population and biomarkers increase level.

You specify that recent studies have shown a correlation between elevated NSE and S100B levels and a risk of cognitive decline and postoperative delirium, see references 8 and 9. What does your study add compared to these two references?

We have revised the concluding part of the introduction, adding the possible future implications of the research.

I don't really understand your statement about the surgeon and his 5 years of experience. Furthermore, were all the procedures actually performed by the same surgeon? Please rephrase/explain and clarify.

We thank the reviewer. The sentence has been rephrased to make it clearer.

You report that your data demonstrated a correlation between advanced age, diabetes, and poorer performance on neuropsychological questionnaires. Is this consistent with the existing literature? Can you correlate these results with the literature? Please clarify and elaborate.

We expanded the discussions by introducing additional references to validate our results.

Excellent discussion; very clear and relevant.

You suggest that NSE and S100B may be valuable biomarkers for assessing the risk of postoperative cognitive decline. What might the strategy be in this regard? What would be your advice or recommendations for the future? Do you recommend systematic screening of these biomarker levels preoperatively and postoperatively to detect neurological decline earlier? What do you know about the cost of these analyses? Could they be systematically added to the preoperative assessment before orthopedic surgery? Before other surgeries? Is this easily achievable by most laboratories? Is the cost worth the analysis?

How can we organize a strategy using these biomarkers to truly reduce neurological decline in clinical practice? Could you explain, rephrase, or provide details in the text?

Thank you.

We have completely rewritten the conclusions of our work, addressing the questions requested. According to our study, this research could open up further scenarios for laboratory diagnostics in patients with femoral neck fractures who present negative prognostic factors at diagnosis, as shown in the logistic regression.

Reviewer 2 Report

Comments and Suggestions for Authors
  • Dear colleagues, some points to improve the paper. Please note none of the suggested references are mine so feel free to add them or reject them. I am also non-English speaker but noticed several minor grammatical issues ("impairment neurological history absence" on line 101; "U Mann-Whitney test" should be "Mann-Whitney U test") pls seek or arrange for English-language Editing.
  • The title includes long term follow-up. 12 months is generally not considered long-term follow-up thus I suggest to replace it with one-year follow-up.
  • Abstract need to explicitly explain that sample drop from 146 to 95 at one-year follow-up.
  • The introduction provides a good overview of POCD pathogenesis but refs 1-10 mostly < 2015. See example: ref 1 Tomaszewski D. Biomarkers of Brain Damage and Postoperative Cognitive Disorders in Orthopedic Patients: An Update. Biomed Res Int. 2015;2015:402959. doi: 10.1155/2015/402959. Epub 2015 Aug 31. PMID: 26417595; PMCID: PMC4568345.
  • It can be updated to newer one see Ntalouka MP, Arnaoutoglou E, Tzimas P. Postoperative cognitive disorders: an update. Hippokratia. 2018 Oct-Dec;22(4):147-154. PMID: 31695301; PMCID: PMC6825421.
  • Or even better Zhao Q, Wan H, Pan H, Xu Y. Postoperative cognitive dysfunction-current research progress. Front Behav Neurosci. 2024 Jan 30;18:1328790. doi: 10.3389/fnbeh.2024.1328790. PMID: 38357422; PMCID: PMC10865506.
  • Or even better Wang, T.; Huang, X.; Sun, S.; Wang, Y.; Han, L.; Zhang, T.; Zhang, T.; Chen, X. Recent Advances in the Mechanisms of Postoperative Neurocognitive Dysfunction: A Narrative Review. Biomedicines 2025, 13, 115. https://doi.org/10.3390/biomedicines13010115
  • Study design is missing power analysis + details on inter-site standardization (surgical protocols or biomarker assay variations) + improved Inclusion/Exclusion Criteria.
  • When discussing the Fracture Classification explain the methods use on how discrepancies between the two independent reviewers/ doctors (G.M. and V.C.) were resolved.
  • For ELISA for S100B and Elecsys for NSE provide intra- and inter-assay coefficients of variation to assess reliability.
  • Ref #16 please check and if necessary update to Foderaro G, Isella V, Mazzone A, Biglia E, Di Gangi M, Pasotti F, Sansotera F, Grobberio M, Raimondi V, Mapelli C, Ferri F, Impagnatiello V, Ferrarese C, Appollonio IM. Brand new norms for a good old test: Northern Italy normative study of MiniMental State Examination. Neurol Sci. 2022 May;43(5):3053-3063. doi: 10.1007/s10072-021-05845-4. Epub 2022 Jan 6. Erratum in: Neurol Sci. 2024 Nov;45(11):5563-5564. doi: 10.1007/s10072-024-07585-7. PMID: 34989910; PMCID: PMC9018649.
  • Pls use Italian notms to justify the cutoff (MMSE <20) with references, as cultural/educational biases are acknowledged but not mitigated. Also 20-11 is usually interpreted as moderate not severe.
  • Study showed 28 excluded pre-surgery for POCD and 51 lost at one-year describe any efforts to minimize loss and statistical handling.
  • Power analysis should be part of methods not in statistical analyses.
  • for logistic regressions pls report model fit Hosmer-Lemeshow test to validate assumptions.
  • Table 4: ORs for age (1.24) diabetes (4.41) MMSE (0.25) and PS (9.81) are reported pls include any multivariate adjustment details and check for multicollinearity between MMSE and PS.
  • The discussion I suggest to expand on potential confounders like postoperative rehabilitation adherence or medication (sleep or pain medicine) use which could influence long-term POCD. See Huang WWY, Fan S, Li WY, Thangavelu V, Saripella A, Englesakis M, Yan E, Chung F. Prevalence of postoperative neurocognitive disorders in older non-cardiac surgical patients: A systematic review and meta-analysis. J Clin Anesth. 2025 Apr;103:111830. doi: 10.1016/j.jclinane.2025.111830. Epub 2025 Apr 7. PMID: 40199029.
  • Minor grammatical issues ("impairment neurological history absence" on line 101; "U Mann-Whitney test" should be "Mann-Whitney U test")
Comments on the Quality of English Language
  • Minor grammatical issues ("impairment neurological history absence" on line 101; "U Mann-Whitney test" should be "Mann-Whitney U test")

Author Response

Semptember 22, 2025

Journal of Functional Morphology and Kinesiology

Reviewers

& c.c.

Manuscript Editor

Dear Editor,

            Thank you for your thoughtful and constructive comments. In this cover letter, we have addressed each of the issues raised and have highlighted the relevant revisions in the manuscript itself (underlined). Below, please find item-by-item responses to the Reviewers’ comments.

Please note: Editor’s and Reviewers’ comments are in italicized red font; Authors’ answers are in regular black font;

Sincerely yours,

Michele Coviello, MD PhD student

Orthopaedics Unit, Department of Clinical and Experimental Medicine, Faculty of Medicine and Surgery, University of Foggia, Policlinico Riuniti di Foggia, 71122 Foggia, Italy

E-mail: michelecoviello91@gmail.com, ORCID: 0000-0003-3585-1000, Phone: +393938165088

Reviewer 2

Dear colleagues, some points to improve the paper. Please note none of the suggested references are mine so feel free to add them or reject them. I am also non-English speaker but noticed several minor grammatical issues ("impairment neurological history absence" on line 101; "U Mann-Whitney test" should be "Mann-Whitney U test") pls seek or arrange for English-language Editing.

The manuscript has been reviewed by native English speaker.

The title includes long term follow-up. 12 months is generally not considered long-term follow-up thus I suggest to replace it with one-year follow-up.

Abstract need to explicitly explain that sample drop from 146 to 95 at one-year follow-up.

We corrected title and abstract according your suggestions.

The introduction provides a good overview of POCD pathogenesis but refs 1-10 mostly < 2015. See example: ref 1 Tomaszewski D. Biomarkers of Brain Damage and Postoperative Cognitive Disorders in Orthopedic Patients: An Update. Biomed Res Int. 2015;2015:402959. doi: 10.1155/2015/402959. Epub 2015 Aug 31. PMID: 26417595; PMCID: PMC4568345.

It can be updated to newer one see Ntalouka MP, Arnaoutoglou E, Tzimas P. Postoperative cognitive disorders: an update. Hippokratia. 2018 Oct-Dec;22(4):147-154. PMID: 31695301; PMCID: PMC6825421.

  • Or even better Zhao Q, Wan H, Pan H, Xu Y. Postoperative cognitive dysfunction-current research progress. Front Behav Neurosci. 2024 Jan 30;18:1328790. doi: 10.3389/fnbeh.2024.1328790. PMID: 38357422; PMCID: PMC10865506.
  • Or even better Wang, T.; Huang, X.; Sun, S.; Wang, Y.; Han, L.; Zhang, T.; Zhang, T.; Chen, X. Recent Advances in the Mechanisms of Postoperative Neurocognitive Dysfunction: A Narrative Review. Biomedicines 2025, 13, 115. https://doi.org/10.3390/biomedicines13010115

We have replaced the references with the most recents as suggested.

Study design is missing power analysis + details on inter-site standardization (surgical protocols or biomarker assay variations) + improved Inclusion/Exclusion Criteria.

When discussing the Fracture Classification explain the methods use on how discrepancies between the two independent reviewers/ doctors (G.M. and V.C.) were resolved.

For ELISA for S100B and Elecsys for NSE provide intra- and inter-assay coefficients of variation to assess reliability.

We improve method section according to suggestions.

Ref #16 please check and if necessary update to Foderaro G, Isella V, Mazzone A, Biglia E, Di Gangi M, Pasotti F, Sansotera F, Grobberio M, Raimondi V, Mapelli C, Ferri F, Impagnatiello V, Ferrarese C, Appollonio IM. Brand new norms for a good old test: Northern Italy normative study of MiniMental State Examination. Neurol Sci. 2022 May;43(5):3053-3063. doi: 10.1007/s10072-021-05845-4. Epub 2022 Jan 6. Erratum in: Neurol Sci. 2024 Nov;45(11):5563-5564. doi: 10.1007/s10072-024-07585-7. PMID: 34989910; PMCID: PMC9018649.

Pls use Italian notms to justify the cutoff (MMSE <20) with references, as cultural/educational biases are acknowledged but not mitigated. Also 20-11 is usually interpreted as moderate not severe.

We replaced the reference including Italian cutoff for the test.

Study showed 28 excluded pre-surgery for POCD and 51 lost at one-year describe any efforts to minimize loss and statistical handling.

Power analysis should be part of methods not in statistical analyses.

We corrected and improved paper according to suggestions.

For logistic regressions pls report model fit Hosmer-Lemeshow test to validate assumptions.

We used Hosmer-Lemeshow test to validate to improve regression model.

Table 4: ORs for age (1.24) diabetes (4.41) MMSE (0.25) and PS (9.81) are reported pls include any multivariate adjustment details and check for multicollinearity between MMSE and PS. ??

We have improved the statistical part of the methods by clarifying some details that are useful for the reader to understand the tests performed. In the results, we explained in depth that the values expressed in the table had already been adjusted. Even the neurological tests themselves were left as individual items in the regression because they had already been tested a priori. We included everything in the manuscript to avoid misunderstandings.

The discussion I suggest to expand on potential confounders like postoperative rehabilitation adherence or medication (sleep or pain medicine) use which could influence long-term POCD. See Huang WWY, Fan S, Li WY, Thangavelu V, Saripella A, Englesakis M, Yan E, Chung F. Prevalence of postoperative neurocognitive disorders in older non-cardiac surgical patients: A systematic review and meta-analysis. J Clin Anesth. 2025 Apr;103:111830. doi: 10.1016/j.jclinane.2025.111830. Epub 2025 Apr 7. PMID: 40199029.

We improve discussion including reference.

Minor grammatical issues ("impairment neurological history absence" on line 101; "U Mann-Whitney test" should be "Mann-Whitney U test")

We have carefully taken all comments and suggestions into consideration and have made thoroughly corresponding revisions in the revised manuscript file.

We thank you for giving us this opportunity.

We hope to continue a collaboration with your Journal.

Round 2

Reviewer 1 Report

Comments and Suggestions for Authors

Hello,
Thank you for this correction.
All comments have been taken into account and have been addressed in a satisfactory and correct manner.
Congratulations.
In my opinion, it can be published in this revised form.
Thanks again.

Reviewer 2 Report

Comments and Suggestions for Authors

no more comments